# Comparative Morphology of the Stinger in Social Wasps (Hymenoptera: Vespidae)

**DOI:** 10.3390/insects12080729

**Published:** 2021-08-14

**Authors:** Mario Bissessarsingh, Christopher K. Starr

**Affiliations:** 1Department of Life Sciences, University of the West Indies, St Augustine, Trinidad and Tobago; nismofan@gmail.com; 2San Fernando East Secondary School, Pleasantville, Trinidad and Tobago

**Keywords:** social wasps, sting autotomy, venom apparatus, Vespidae

## Abstract

**Simple Summary:**

Both solitary and social wasps have a fully functional venom apparatus and can deliver painful stings, which they do in self-defense. However, solitary wasps sting in subduing prey, while social wasps do so in defense of the colony. The structure of the stinger is remarkably uniform across the large family that comprises both solitary and social species. The most notable source of variation is in the number and strength of barbs at the tips of the slender sting lancets that penetrate the wound in stinging. These are more numerous and robust in New World social species with very large colonies, so that in stinging human skin they often cannot be withdrawn, leading to sting autotomy, which is fatal to the wasp. This phenomenon is well-known from honey bees.

**Abstract:**

The physical features of the stinger are compared in 51 species of vespid wasps: 4 eumenines and zethines, 2 stenogastrines, 16 independent-founding polistines, 13 swarm-founding New World polistines, and 16 vespines. The overall structure of the stinger is remarkably uniform within the family. Although the wasps show a broad range in body size and social habits, the central part of the venom-delivery apparatus—the sting shaft—varies only to a modest extent in length relative to overall body size. What variation there is shows no apparent correlation with social habits. This is consistent with the hypothesis that stinger size is constrained by the demands of a flight-worthy body. The sting lancets bear distinct, acute barbs in all examined species except in members of the Stenogastrinae. Barbs vary considerably among species in number, their summed lengths, and the relative degree of serration (summed length relative to lancet width). Where they are numerous and strong, it increases the likelihood of the stinger remaining fatally embedded in the skin of a vertebrate adversary (sting autotomy). Although an index that combines the number and strength of barbs is a more natural measure of overall serration, the number of barbs alone is almost as good a predictor of the likelihood of sting autotomy. Across the family as a whole, the tendency to sting autotomy is concentrated in the swarm-founding New World polistines.

## 1. Introduction

The venom apparatus is the outstanding shared derived character of the aculeate hymenoptera, the large group that includes wasps, bees, and ants. It comprises a glandular–muscular component and a cuticular complex, the latter known as the sting or stinger [1] (Figure 1). Primitively, the venom apparatus serves in subduing prey, but in several lineages it has taken on the important secondary function of defense of self and brood. In some of these lineages, stinging no longer serves to subdue prey but is entirely defensive.

After its synthesis in specialized glands, the venom is held in the venom sac (venom reservoir) until it is expressed into the wound in the act of stinging or spraying. In some lineages this is accomplished by means of muscles around the venom reservoir (the injection type of, e.g., Pompilidae and Vespidae), while in some others it is by means of valves on the sting lancets (the valve-pump type of, e.g., Apoidea and Formicidae) [3].

There are various ways that the venom apparatus can be enhanced as a more effective defensive device or, alternatively, reduced in structure and function [4,5]. Our focus here is on two parts of the stinger. In aculeates, the paired first valvulae, or lower valves, are long, fine needles, or lancets ((Figure 6 of [1]), [6,7]), that curve down and posteriorly to the tip of the stinger. The second valvulae, or upper valves, are fused dorsally in most aculeates to form the much broader sting shaft, which envelops the lancets and within which they move posterior–anteriorly in the course of stinging. These two components can be regarded as the “business end” of the venom apparatus, which penetrate the prey/victim, while all other components are servo-mechanisms to their functioning. The length of the sting shaft relative to overall body size provides a convenient index of relative physical investment in the venom apparatus.

The tip of each lancet often has a series of distinct, anterior-directed (i.e., away from the lancet tip) barbs. In his pioneering microanatomical study of the honey bee *Apis mellifera*, Jan Swammerdam ([8] page 199, Table XVIII) clearly figured the lancet barbs (Figure 2) and related them to the well-known phenomenon of sting autotomy. In this, the stinger becomes irrecoverably, fatally embedded in the skin when stinging a human [9,10]. Snodgrass [11] attributed to the barbs an anchoring function as the lancets in turn penetrate deeper into the wound. The venom apparatus has been studied in the various species of honey bees (*Apis* spp.), all virtually identical in overall structure with 10–11 acute barbs per lancet [12].

Tautz [13] noted that a honey bee can withdraw the stinger after stinging an arthropod and suggested that the inability to do so when stinging a vertebrate may be seen instead as “an evolutionary ‘mistake’ on the bees’ part.” However, the prevailing opinion follows Charles Darwin’s ([14] Chapter 6) view of sting autotomy as a feature favored by natural selection in the interest of the colony as a whole and therefore in no way maladaptive. 

The micro-anatomist Léon Dufour may have been the first to appreciate that lancet barbs are not peculiar to honey bees but widespread in the Aculeata [15]. In a comparative treatment of the anatomy of the venom apparatus, Rietschel (Figures 14, 16, and 17 of [16]) illustrated the tips of sting lancets in five species of social wasps and several other hymenoptera.

A study of 102 species of wasps and bees (Table 1 of [17]) reported at least one barb (not always acute) on the lancets of all but one species. Among the Vespidae, these included 14 species of solitary eumenines (most with 4 or 5 barbs per lancet), 8 independent-founding polistines (most commonly with 5 barbs), 1 swarm-founding polistine (7 or 8 barbs), and 4 vespines (8 or 9 barbs). Consistent with these findings, a study across the families of bees [6] reported the presence of barbs in most solitary bees and all social bees in which the stinger as a whole is not greatly reduced. As a rule, spheciform wasps have a small number of distinct but relatively weak barbs ([6,17], (Figures 63–65 of [18]), (Plates 29–31 of [19]); personal observation). In contrast, all studied species of spider wasps (Pompilidae) have smooth lancets without barbs ([20]; D. Gladun, personal communication; personal observation). The presence or absence of barbs is very variable across the family of ants ([20,21,22]; Table 2 of [23]). In addition, similar barbs are present on the first valvulae of many parasitic wasps (D.J. Brothers, personal communication; D.J. Quicke, personal communication [16]). Accordingly, the presence of lancet barbs appears to be in the ground plan of the Aculeata.

Macalintal and Starr [2] combined the number and size of lancet barbs into a single serration index, Sr = Σb_i_/h, in which b_i_ is the length of the *i*th barb along its anterior edge and h is the width of the lancet in midway along its final, straight section. 

The family Vespidae comprises an estimated 4000 known species in eight subfamilies [24]. Of the five subfamilies of solitary wasps, the two largest (Eumeninae and Zethinae) are treated here [25]. The other three subfamilies are social wasps, characterized by living in durable structured groups (colonies). These latter show considerable diversity in colony composition. The Stenogastrinae and the greater part of the Polistinae are independent founders whose relatively small colonies are each founded by one or several queens without the aid of workers [26,27]. A large minority of polistine species are swarm founders, whose often very large colonies are founded by groups of many queens and workers together [28]. In most Vespinae, unlike the other two subfamilies, queens and workers are physically clearly distinct, at least in size [29]. The Polistinae comprise four tribes, the Ropalidiini, Mischocyttarini (*Mischocyttarus*), Polistini *(Polistes*), and the New World swarm-founding Epiponini. Accordingly, our treatment embraces eight non-nested natural groups.

There is consensus about the phylogenetic relations among these eight groups except on one point. The traditional scheme, supported by a recent analysis [30], places the Stenogastrinae as the sister-group of the Polistineae and Vespinae, while an alternative recent analysis [31] shows them as the sister-group of all other subfamilies.

Against this broad social diversity, vespids show only modest physical diversity. In particular for our purposes, all studied species show a near uniformity in the gross structure of the venom apparatus, such that differences appear only in the relative sizes of some components [2,32,33,34,35,36]. While solitary vespids appear usually to sting their prey into paralysis, present indications are that social species sting in defense of self or colony but seldom in subduing prey.

To date there have been two main comparative studies of stinger structure in vespid wasps. Macalintal and Starr [2] examined the stinger of 39 species of the Old World social genus *Ropalidia*, of which 21 were known or believed to be independent-founding, 9 known or believed to be swarm-founders, and 9 regarded as intermediate in habits. They compared (a) the length of the furcula, a small sclerite with a key role in manipulating the sting shaft [37], (b) the length of the sting shaft relative to body size, and (c) the serration index. Against their prediction, they found no correlation between social habit and any of these measures. Although occasional sting autotomy had been noted in some swarm-founding *Ropalidia*, the data are too sparse to draw conclusions.

In the other major comparative study, Manzolini-Palma and Gobbi [35] studied the lancets of 8 independent-founding and 20 swarm-founding neotropical polistine wasps. Their nine plates of elegant photomicrographs of the lancet tips show the barbs with exceptional clarity. They also induced attack against a target with simulated skin in order to record the presence or absence of sting autotomy. Among their results, autotomizing species tended to have more barbs than those that did not autotomize.

With these two studies as background, we broaden the taxonomic scope by treating representatives of four subfamilies of vespids with different social habits. In these, we record (a) the ratio of sting-shaft length to forewing length as an index of relative investment in the venom apparatus, and (b) not only the number of lancet barbs but their indices of serration. It is predicted that these will show significant differences in line with social differences, such that taxa with larger colonies and more to defend will have (a) larger stingers, and (b) a greater tendency to sting autotomy. The logic of the second prediction, following Darwin, is that sting autotomy, while fatal to the stinging insect, can benefit the colony as a whole through more effective defense [38].

## 2. Materials and Methods

Our study material comprised representatives of all three social subfamilies and the two largest solitary subfamilies of Vespidae, with attention to the sting lancets of honey bee *Apis mellifera* for comparison. Our methods follow Macalintal and Starr (1996) [2]. Forewing length and head width are among the common indices of overall body size in social wasps, e.g., [39]. We measured the first to the nearest half-millimeter with an ordinary ruler, the second to the nearest 0.06 mm by means of an eyepiece micrometer in a dissecting microscope. 

We extracted stingers from freshly killed wasps or dry specimens that had been relaxed, and cleared them overnight in dilute KOH. We then microphotographed the sting shaft and the distal, straight part of the sting lancets at known magnifications. From photographs we took four measures from each specimen: (a) length of the sting shaft, (b) width of one of the lancets about midway along the straight part, (c) number of lancet barbs, and (d) summed length of the lancet barbs. 

In contrast to Manzolini-Palma and Gobbi [35], we made no attempt at a controlled experimental record of sting autotomy. Rather, our data are drawn from our own personal field experience and that elicited from colleagues.

In conducting statistical tests among groups of species, we made no attempt to include all pairs of sister-groups, as our focus was on social comparisons and only tangentially on phylogenetic aspects. Accordingly, some paraphyletic groups (e.g., all primitively social species) are included. 

## 3. Results

Forewing length seemed more consistent than head width as an index of body size. That is, some wasps appeared to have uncommonly broad or narrow heads relative to overall body size. The ratio of sting-shaft length to wing length, therefore, is taken here as an index of relative physical investment in the stinger. This index shows only modest variation across the family; in almost three-quarters of the species studied, it was between 0.17 and 0.21 (Table 1). Nonetheless tests of difference of mean show a significance difference in all but one of the comparisons between groups (Table 2).

Unlike stinger size relative to forewing length, there is wide variation in number of barbs and serration, with a strong phylogenetic component to the variation (Table 2). Studied species of the solitary eumenines and zethines have fewer barbs and lower serration than any polistine or vespine (Table 1, Figure 3a,b). On the other hand, the two studied species of the primitively social stenogastrines have no barbs at all (Figure 3c,d). Among polistines, there is a clear dichotomy between the independent-founding and New World swarm-founding genera. The former have moderate serration (Figure 3e,f), and sting autotomy is not known in any of them. Most of the latter have distinctly higher serration (Figure 3g–m), and autotomy is known from many of them. *Epipona tatua*, for example, closely resembles *Apis mellifera* in number of barbs and relative serration, so that its autotomy was predictable. Studied vespines have moderate serration (Figure 3n–p) distinctly lower than that of some swarm-founders.

As expected, the serration index *Sr* is closely related to the number of barbs on the sting lancets. These two parameters likewise show a positive correlation with available data on sting autotomy in vertebrate skin, although with some anomalies (Table 1). In particular, some vespines with many strong barbs appear usually not to autotomize, possibly because these robust wasps have the physical strength to tear the lancets free of the wound.

Combining those species in which sting autotomy is recorded as occasional (±) and usual (+) vs. those in which it is rare or unknown (−) (Table 1), yields a distinct positive relationship between sting autotomy and each of three lancet parameters: (a) number of barbs, (b) absolute serration, and (c) the serration index (Mann–Whitney U one-tailed test, *p* < 0.01). 

## 4. Discussion

The similarity within relatively limits across the Vespidae of sting-shaft length relative to forewing length speaks against the prediction that species with larger colonies and therefore more to defend will tend to have greater investment in the venom apparatus. It speaks instead for the hypothesis that the size of the venom apparatus is constrained by the demands of flight. This is consistent with the more general thesis that body forms are constrained by a narrow range of workable solutions to environmental challenges [40], as well as the more specialized idea that the demands of flight may prevent the appearance of extreme caste polymorphism in social wasps and bees, unlike in ants and termites [41].

Where they are a replication, our data corroborate those of Manzolini-Palma and Gobbi [35]. In particular, the numbers of lancet barbs in different species are almost identical in their results and ours. In addition, our sting-autotomy results from field experience match their experimental results.

The data show a strong positive correlation between sting lancet serration and the incidence of sting autotomy in human skin. There is an approximate dividing line between species with relative serration Sr less than 1 (autotomy very unlikely) and those above 1 (autotomy likely at least some of the time). However, the number of barbs per lancet alone is almost as reliable an index, such that autotomy is not expected where there are fewer than eight barbs, and common or almost certain with eight or more barbs. On this basis, among the species in Table 1, sting autotomy at least some of the time is predicted in *Vespula*
*consobrina* and *V. squamosa*.

## 5. Conclusions

In all social wasps and their close solitary relatives, the females have the apparatus and ability needed to inflict painful stings. We find a remarkable uniformity in the structure of the venom apparatus across the broad diversity of social patterns in the family. The various groups show modest but statistically significant differences in physical investment in the venom apparatus. More striking are differences in the number and strength of barbs on the fine lancets that penetrate the skin in the course of stinging. Where these are many and large, it can lead to sting autotomy, the fatal inability to withdraw the stinger from the (vertebrate) adversary’s skin. The degree of development of this particular aspect of the stinger is associated with social patterns of the species.

## Figures and Tables

**Figure 1 insects-12-00729-f001:**
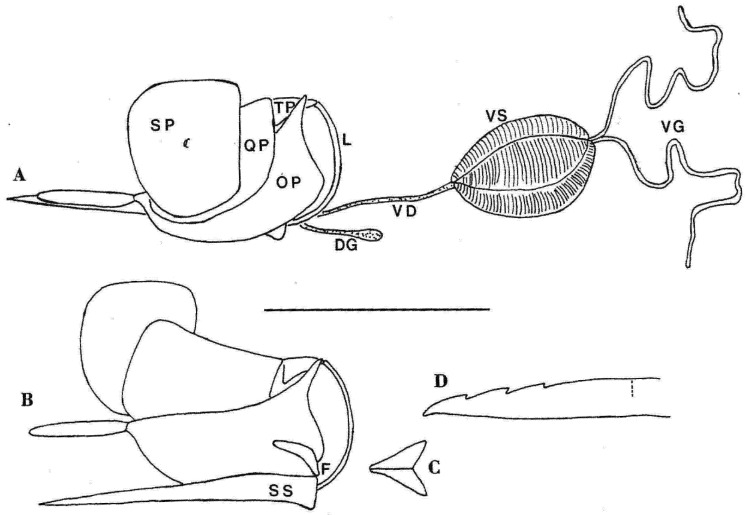
Venom apparatus of *Ropalidia horni* to show features common to the Vespidae. (**A**) Side view from the left of the intact venom apparatus with soft tissues removed except for the muscles surrounding the venom sac. (**B**) Side view from the left of the stinger, with components in place but splayed. (**C**) Furcula in top view. (**D**) Tip of a lancet. Scale bar = 1 mm for (**A**–**C**), approximately 0.1 mm for (**D**). DG = Dufour’s gland. F = furcula. OP = oblong plate. QP = quadrangular plate. SP = spiracular plate. SS = sting shaft. TP = triangular plate. VD = venom duct. VG = venom gland. VS = venom sac. From Macalintal and Starr [2].

**Figure 2 insects-12-00729-f002:**
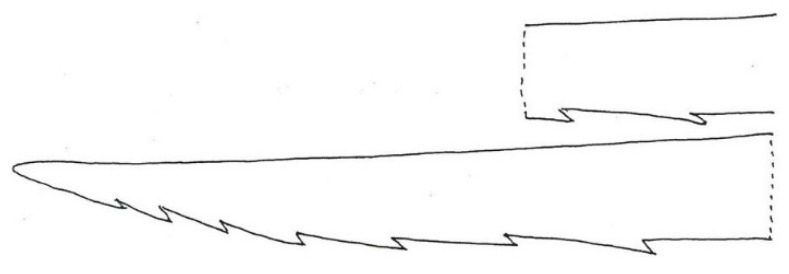
Terminal part of a sting lancet in the western honey *Apis mellifera* bee to illustrate the number and strength of the barbs. It has a very high serration index (defined in the text) of Sr = 2.43. Width at the dashed line is approximately 0.04 mm.

**Figure 3 insects-12-00729-f003:**
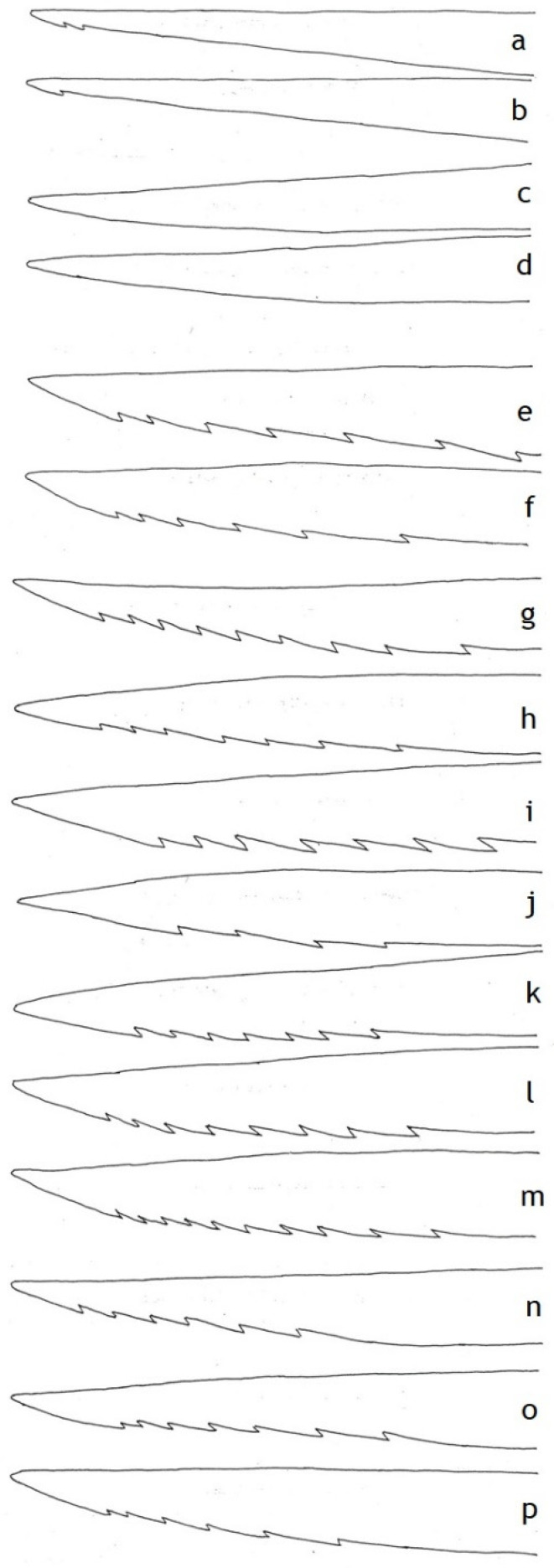
Terminal part of a sting lancet in vespid wasps in top or bottom view to illustrate the number and strength of barbs. Not all to the same scale. The bracketed number after each name is the serration index (*Sr*) defined in the text. (**a**) *Pachodynerus* sp. B (0.10). (**b**) *Zethus* sp. A (0.11). (**c**) *Eustenogaster*
*calyptodoma* (0). (**d**) *Parischnogaster striatula* (0). (**e**) *Polistes lanio* (0.65). (**f**) *Polistes versicolor* (0.76). (**g**) *Agelaia centralis* (1.66). (**h**) *Angiopolybia pallens* (0.68). (**i**) *Epipona tatua* (2.27). (**j**) *Parachartergus*
*fraternus* (0.54). (**k**) *Polybia occidentalis* (1.20). (**l**) *Polybia rejecta* (1.17). (**m**) *Synoeca surinama* (1.59). (**n**) *Dolichovespula maculata* (0.76). (**o**) *Vespa bicolor* (0.84). (**p**) *Vespa ducalis* (0.79).

**Table 1 insects-12-00729-t001:** Features of the stinger in vespid wasps and the honey bee *Apis mellifera*. In the Vespinae and *Apis mellifera*, all specimens are apparent workers. The bracketed number after each species name is the sample size. All figures are means, except number of barbs, which is the mode. Only barbs forming an acute angle proximally are counted. Absolute serration is the summed depths of all barbs, while the serration index is the ratio of absolute serration to the height of the sting lancet. Sting autotomy in human skin is coded as almost never (−), often (±), or usually (+) occurring according to literature reports and many personal communications; signs in square brackets are inferred from closely related species.

	Shaft Length (mm)	Wing Length (mm)	Shaft/Wing ± SD	Head Width (mm)	Shaft/Head	Lancet Barbs	Absolute Serration	Serration Index *Sr*	Sting Autotomy
					(μm)				
EUMENINAE and ZETHINAE									
*Pachodynerus* sp. A (3)	2.99	14.5	0.21 ± 0.01	4.18	0.72	3	11	0.22 + 0.02	
*Pachodynerus* sp. B (5)	2.81	13.6	0.21 ± 0.01	3.86	0.73	2	5	0.10 + 0.02	
*Zethus* sp. A (1)	1.76	10.5	0.17	3.96	0.45	2	4	0.14	
*Zethus* sp. B (3)	1.67	9.2	0.18 ± 0.02	3.77	0.44	1	5	0.08 + 0.02	
STENOGASTRINAE									
*Eustenogaster calyptodoma* (5)	3.01	12.0	0.25 ± 0.01	3.71	0.81	0	0	0	-
*Parischnogaster striatula* (4)	1.57	8.8	0.18 ± 0.01	2.68	0.59	0	0	0	
POLISTINAE (Independent-Founding)									
*Belonogaster juncea* (5)	2.29	18.6	0.12 ± 0.01	3.82	0.60	6	47	0.62 + 0.06	-
*Mischocyttarus alfkeni* (5)	1.34	9.8	0.14 ± 0.01	2.58	0.52	4	33	0.53 + 0.04	-
*Mischocyttarus labiatus* (5)	1.43	14.7	0.10 + 0.01	3.31	0.44	5	31	0.66 + 0.06	-
*Parapolybia indica* (3)	2.70	16.0	0.17 + 0.07	3.86	0.70	6	53	0.77 + 0.22	-
*Parapolybia varia* (5)	1.90	10.9	0.17 + 0.07	3.09	0.62	6	42	0.87 + 0.25	-
*Polistes dominula* (5)	2.37	12.4	0.19 + 0.00	3.75	0.63	5	23	0.40 + 0.05	-
*Polistes dorsalis* (5)	2.01	11.3	0.18 + 0.01	3.19	0.63	5	30	0.55 + 0.08	-
*Polistes fuscatus* (2)	2.52	14.5	0.17 + 0.01	4.02	0.63	6	37	0.54 + 0.00	-
*Polistes gigas* (4)	5.42	26.0	0.21 + 0.00	6.50	0.84	7	84	0.64 + 0.13	
*Polistes lanio* (5)	3.26	19.1	0.17 + 0.01	4.41	0.74	6	49	0.65 + 0.05	-
*Polistes metricus* (4)	2.75	15.3	0.18 + 0.01	4.30	0.64	5	37	0.55 + 0.02	-
*Polistes olivaceus* (5)	3.05	15.4	0.20 + 0.01	4.48	0.68	6–7	50	0.64 + 0.18	
*Polistes stigma* (8)	2.19	9.4	0.23 + 0.01	3.22	0.68	4	26	0.44 + 0.14	
*Polistes strigosus* (1)	3.88	20.0	0.19	4.82	0.80	5	46	0.54	
*Polistes tenebricosus* (2)	3.77	20.5	0.19 + 0.01	4.90	0.77	6	59	0.64 + 0.09	
*Polistes versicolor* (5)	2.49	13.2	0.19 + 0.01	3.19	0.78	6	59	1.03 + 0.17	-
POLISTINAE (Swarm-Founding)									
*Agelaia centralis* (4)	1.23	9.5	0.13 + 0.01	2.73	0.45	9	100	1.66 + 0.23	+
*Angiopolybia pallens* (5)	1.12	7.5	0.15 + 0.01	2.15	0.52	6	25	0.68 + 0.08	-
*Apoica pallens* (5)	1.91	14.6	0.13 + 0.01	3.24	0.59	6	63	0.91 + 0.10	-
*Apoica pallida* (5)	2.21	16.1	0.14 + 0.01	3.40	0.65	8–9	75	1.14 + 0.03	±
*Brachygastra bilineolata* (5)	1.60	5.3	0.31 + 0.04	2.54	0.63	6	47	0.93 + 0.09	+
*Epipona tatua* (5)	1.46	10.2	0.14 + 0.00	3.73	0.39	9	124	2.27 + 0.42	+
*Metapolybia cingulata* (5)	1.05	5.3	0.20 + 0.03	5.25	0.47	7	48	0.82 + 0.04	±
*Parachartergus colobopterus* (5)	1.26	6.7	0.19 + 0.01	2.36	0.53	4	20	0.43 + 0.12	-
*Parachartergus fraternus* (5)	2.11	11.2	0.19 + 0.01	3.22	0.66	4	34	0.54 + 0.11	±
*Polybia occidentalis* (5)	0.42	6.0	0.07 + 0.07	2.00	0.21	8	54	1.20 + 0.10	±
*Polybia rejecta* (5)	1.40	7.3	0.19 + 0.01	2.55	0.55	8	82	1.17 + 0.07	±
*Protopolybia exigua* (4)	0.87	3.9	0.23 + 0.01	1.50	0.58	6	18	0.64 + 0.05	+
*Synoeca surinama* (5)	3.00	18.9	0.16 + 0.01	4.84	0.62	12	213	1.59 + 0.37	+
VESPINAE									
*Dolichovespula arenaria* (5)	1.95	9.4	0.21 + 0.00	3.50	0.56	6	48	0.61 + 0.04	-
*Dolichovespula maculata* (5)	2.61	13.9	0.19 + 0.01	4.72	0.56	7	68	0.76 + 0.03	-
*Dolichovespula norvegicoides* (4)	1.85	9.6	0.19 + 0.01	3.40	0.54	6–7	51	0.73 + 0.07	-
*Vespa affinis* (4)	3.16	17.6	0.18 + 0.00	5.72	0.55	7	68	0.72 + 0.06	-
*Vespa bicolor* (5)	2.91	13.9	0.21 + 0.01	5.17	0.57	7	82	0.84 + 0.08	-
*Vespa crabro* (5)	3.60	19.1	0.19 + 0.01	6.25	0.58	7	68	0.68 + 0.07	-
*Vespa ducalis* (5)	4.25	25.5	0.17 + 0.01	7.45	0.57	6	98	0.75 + 0.26	-
*Vespa luctuosa* (5)	2.97	13.8	0.20 + 0.01				7		0.75 + 0.07
*Vespa tropica* (5)	3.71	21.3	0.17 + 0.00	6.16	0.59	6–7	82	0.81 + 0.10	-
*Vespa velutina* (4)	3.20	17.3	0.19 + 0.00	5.66	0.56	7	79	0.80 + 0.11	-
*Vespula atropilosa* (4)	2.24	10.8	0.21 + 0.01	3.92	0.57	9	70	1.09 + 0.19	+
*Vespula consobrina* (5)	1.94	9.8	0.20 + 0.01	3.54	0.55	8	57	0.83 + 0.07	
*Vespula germanica* (6)	1.92	9.5	0.20 + 0.01	3.75	0.51	8	69	0.99 + 0.06	±
*Vespula maculifrons* (7)	1.97	10.1	0.19 + 0.00	3.40	0.58	8–9	70	1.15 + 0.18	±
*Vespula pensylvanica* (6)	1.99	10.1	0.20 + 0.01	3.59	0.53	8	74	1.04 + 0.18	±
*Vespula squamosa* (5)	2.16	9.9	0.22 + 0.01	3.60	0.60	9	84	1.30 + 0.19	
*Vespula vidua* (5)	2.23	10.6	0.21 + 0.01	3.78	0.59	8–9	75	1.16 + 0.17	+
*Apis mellifera* (4)	1.81	8.6	0.21 + 0.01	3.48	0.52	8–9	102	2.43 + 0.48	+

**Table 2 insects-12-00729-t002:** Tests (Mann–Whitney U) of difference in means of two key ratios in selected monophyletic groups and paraphyletic groups representing social grades in vespid wasps—based on the same data-set as Table 1. The first ratio under each pair is the relative lengths of the sting shaft and forewing, while the second is the relative serration index. Paraphyletic groups are marked with asterisks. In each comparison, z is the test statistic.

Stenogastrinae vs. Polistinae + Vespinae [30]				
	shaft/forewing	*z* = 2.33, *p* = 0.02	Sr	z = 5.07, *p* < 0.01
Stenogastinae vs. other vespids [31]				
	shaft/forewing	*z* = 2.29, *p* = 0.02	Sr	z = 5.08, *p* < 0.01
Polistinae vs. Vespinae				
	shaft/forewing	*z* = 4.85, *p* < 0.01	Sr	z = 3.19, *p* < 0.01
Polistini vs. Epiponini				
	shaft/forewing	*z* = 3.53, *p* = 0.02	Sr	z = 5.80, *p* < 0.01
Ropalidiini vs. other polistines				
	shaft/forewing	*z* = 2.34, *p* = 0.02	Sr	z = 0.10, *p* = 0.92
solitary wasps * vs. primitively social wasps *				
	shaft/forewing	*z* = 2.23, *p* = 0.03	Sr	z = 4.12, *p* < 0.01
primitively social wasps * vs. Epiponini				
	shaft/forewing	*z* = 1.82, *p* = 0.07	Sr	z = 6.83, *p* < 0.01
primitively social wasps * vs. Vespinae				
	shaft/forewing	*z* = 3.09, *p* < 0.01	Sr	z = 7.75, *p* < 0.01

## Data Availability

Data not archived.

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
