# Peer review of "Comparative Morphology of the Stinger in Social Wasps (Hymenoptera: Vespidae)"

_insects, 2021, doi:10.3390/insects12080729_

Round 1
Reviewer 1 Report
After revision, the manuscript is good enought to be buplished.
I have two minor remarks only:
1) "sting autotomy" is extensively used throughout the text; in lines 18 and 75 it is written in italics, I presume because it is the first time it is used in the Simple Summary and in the main text; if so, it should be written in italics also in the Abstract (line 32).
2) I do not understan line 318; I presume that some word has gone lost in the cut/paste process. Please check
Author Response
Right, sting autotomy italicized in line 32.
A couple of words added to make the suggestion starting in line 318 (now 320) plainer. It's all just about brute force, nothing subtle.
Reviewer 2 Report
Despite new statistical test being performed, they are not described in the methods. Please add a section describing the methods you performed and how you chose the groups for comparison. One concern is that these types of comparisons need to be accompanied by a phylogeny to account for relatedness in any similarities or differences. The alternative is comparing sister pairs throughout the phylogeny. At the very least, this caveat to the conclusions drawn needs to be discussed. Comments indicated in the attached.
Author Response
A paragraph added (lines 184-187) to explain how we chose groups of species for statistical comparison of those two ratios. Given the purposes of the paper, we thought it would have been tedious and rather distracting to treat all pairs of sister groups, especially given the controversial placement of the Stenogastrinae.
This manuscript is a resubmission of an earlier submission. The following is a list of the peer review reports and author responses from that submission.
Round 1
Reviewer 1 Report
The manuscript provided interesting information on stinger morphology in 51 species of social wasps often for the fist time. It discuss the results in relation with autotomy, an important feature of colony defensive behaviour. The paper il well written and only a few minor changes are suggested.
line 57: add L = lancet.
line 58: VS instead of vs.
line 67: "in the sense that it is they" is a bit redundant; please simplify
line 80: why is the lancet divided into to pieces?
line 147: I do not understand what "that if these with show significant differences ..." means; maybe the corret sentence coul be: that if these will show significant differences ...
line 150: whole instead of while
line 159: I do not understand what "relaxed wasps" are; do you mean prepared or pinned dry specimens?
line 162: I would prefere "at half lenghth" instead of "along the middle part"
Table 1: I suggest to repeat the headings also in the second part of the table, thus moving Polistes versicolor in the first half. From the text I understand that the column "Lancet barbs" indicates the number of the barbs, not their lenght; if so (µm) is wrong.
Author Response
I'm afraid I don't see how to call up the reviewer's list of criticisms and suggested changes, so I I will characterize them in turn in responding.
We have made some modest changes in the description of methods, mostly clarifying or elaborating on details.
The reviewer's main criticism had to do with the statistical treatment, which I have expanded and modified. Note that in Table 1 we have added standard deviations only for the two key parameters, as adding them for everything else would seem only to add clutter.
Note that Table 1 is placed at the end, not in its natural place in the text, as it is in landscape orientation, and I had no success inserting it into place without disturbing the overall format.
The reviewer suggested repeating the headings in the second part of the table, but I frankly didn't see what was suggested. I have repeated them at the end, but the production people should feel free to change this if it will make for better layout.
Table 2 is an addition to present the results of tests for differences in means for the key pairwise comparisons of monophyletic or paraphyletic groups. Added text and changes in the results and discussion arise from this table. This is the main revision of the paper.
The reviewer's other minor suggested changes have been noted and, we believe, implemented.
Reviewer 2 Report
This manuscript is a novel contribution to the study of the sting apparatus in solitary and social vespids. I think the study has lots of potential to be highly impactful and robust, but this can only be achieved if the statistical methods are improved. Please see my attached comments.

Author Response

(The authors gave the same response as above.)
